# Position: Ideas Should be the Center of Machine Learning Research

**Jairo Diaz-Rodriguez** [1]

## Abstract

Machine learning research increasingly bifurcates into two disconnected modes: benchmark-driven engineering that prioritizes metrics over understanding, and idealized theory that often fails to transfer to modern systems. In this position paper, we argue that the field focuses too heavily on these endpoints, neglecting the central scientific object: the idea. We propose an Ideas First framework in which *ideas* are valued for the behavioral *signatures* they predict in modern models, and these signatures are tested through *tailored experiments* designed to detect the relevant patterns rather than to win leaderboards. This shift not only bridges the gap between theory and practice but also promotes equity by removing the "complexity premium," enabling rigorous scientific contributions from researchers with modest computational, financial, and human resources. Ultimately, we advocate for a research culture centered on ideas, treating benchmarks and theorems as instruments for testing mechanistic hypotheses rather than as ends in themselves.

## 1. Introduction

Machine learning research has converged on two dominant modes. On the empirical side, shared benchmarks and leaderboards define success: contributions are judged primarily by their effect on a single number on a widely recognized test set (Lipton & Steinhardt, 2018; Dehghani et al., 2021). On the theoretical side, many results are evaluated by the strength of guarantees in highly idealized settings that only partially resemble contemporary overparameterized models (Baraniuk et al., 2020; Zhang et al., 2017). Both modes have delivered real progress, yet they also narrow what counts as a contribution and who can participate, partic-

ularly when large complex models, and extensive ablations become de facto requirements (Ahmed & Wahed, 2020).

**Our position.** In this paper we argue that the central scientific object in machine learning is not a benchmark score or a theorem, but an *idea*: a hypothesis about how learning systems work, what kinds of structure they can exploit, or how they should be evaluated. We call our perspective *Ideas First*. Ideas gain scientific content when they give rise to *signatures* that we can look for in the behavior of modern models (Figure 1), such as specific patterns, characteristic failure modes, or qualitative changes in representation geometry. Experiments are then designed to search for these signatures and rule out competing explanations, rather than primarily to maximize performance on a fixed leaderboard. This reverses the usual order of justification: instead of starting from a large system, benchmark, or theory and asking what results can be extracted, we start from the idea and ask what behavior it predicts and how those predictions can be tested. On this view, simple models, small-scale experiments, and minimal theoretical analyses can be especially valuable when they isolate the mechanism of interest, while large benchmarks, complex systems, and sophisticated theorems remain important when they clarify, operationalize, or stress-test the idea. We are not against benchmarks or theory; we are in favor of placing ideas at the center so that strong ideas can later generate both informative benchmarks and illuminating theorems without being defined by them at birth.

**Equity and simplicity.** This shift also puts equity and simplicity at its core. When publication standards implicitly demand large models, complex implementations, and exhaustive ablations, researchers with substantial computational and engineering resources are favored, while simple but sharp ideas that can be tested in modest settings struggle to be seen (Ahmed & Wahed, 2020; Strubell et al., 2019; Schwartz et al., 2020). Large labs can distribute work across many people, combining teams that run large scale experiments with teams that derive sophisticated theoretical results. This capacity can and does lead to impressive contributions. The problem arises when the same expectations are used as a gatekeeper: an idea can be rejected because it lacks a strong theorem in a highly idealized setting whose assumptions limit its relevance, or because it lacks large benchmark

[1]Department of Mathematics and Statistics, York University, Toronto M3J 1P3, Canada. Correspondence to: Jairo Diaz-Rodriguez <jdiazrod@yorku.ca>.

*Proceedings of the 43ʳᵈ International Conference on Machine Learning*, Seoul, South Korea. PMLR 306, 2026. Copyright 2026 by the author(s).

results that may themselves be vulnerable to overfitting and design artifacts (Dehghani et al., 2021; Bouthillier et al., 2021). History and sociology of science suggest that major advances often emerge from chains of small, conceptually clear steps rather than isolated breakthroughs (Merton, 1973; Kuhn, 1962). A culture that filters out such steps because they are not wrapped in heavy systems or frontier benchmarks risks discarding the very ideas that future theory and benchmarks could build on.

**Change of culture.** Our position is a call to return to hypothesis driven inquiry in the sense used in empirical sciences such as physics and biology. This style of work already underlies much of the most insightful machine learning research (see Section 4), but it is rarely foregrounded in how contributions are framed, evaluated, and rewarded. We argue that the community should normalize the Ideas First perspective, especially for junior researchers, and align reviewing and publishing norms with it. Doing so would deliberately shift incentives away from a status quo that unconsciously mimics the resource intensive priorities of large industrial labs toward a culture in which clear ideas, explicit signatures, and accessible experiments are recognized as core scientific contributions.

**Goals.** The objectives of this position paper are threefold: (i) We propose a framework that links *ideas* to *predicted signatures* to *tailored experiments* for modern neural systems. (ii) We develop a case in favor of idea centric evaluation, connecting it to work on benchmarks, theory, philosophy of science, and equity in AI. (iii) We provide concrete examples of this framework in practice, together with practical notions for evaluating ideas.

**Structure of the paper.** Section 2 introduces our critique of benchmark driven engineering and idealized theory. Section 3 presents the Ideas First framework. Section 4 revisits several deep learning results through this lens, and Section 5 offers practical guidance for authors and reviewers. Section 6 gives a hypothetical case study that illustrates the full workflow. Section 7 develops the broader case for idea centric evaluation and its implications. Finally, we present alternative views in Section 8.

## 2. Standard modes

Before introducing our framework, we briefly characterize two standard modes that structure much of contemporary work on neural systems. While we acknowledge that research methodologies are diverse, we argue that with sufficient flexibility, the incentives of the field tend to align the vast majority of current work with one of these two dominant archetypes.

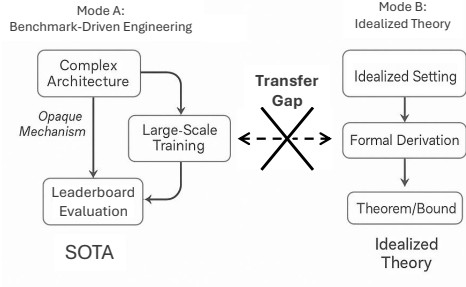

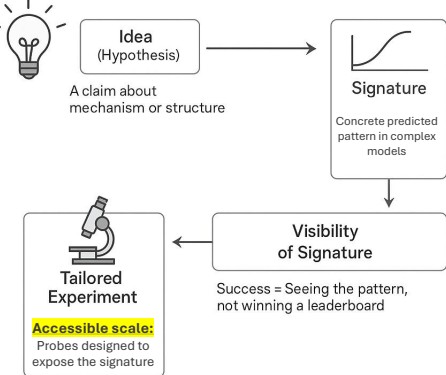

*Figure 1.* (Top) Current practice often bifurcates into *Benchmark-Driven Engineering* (optimizing a single metric on large systems, often obscuring mechanism) and *Idealized Theory* (rigorous proofs in simplified settings that may not transfer to modern models). (Bottom) Our proposed framework links these worlds: An *Idea* generates a concrete *Signature* (an observable behavioral commitment ). A Tailored *Experiment* is then designed specifically to detect this signature, prioritizing mechanistic clarity and accessible experimental design over state-of-the-art leaderboard performance.

### Mode A: Benchmark-driven engineering

This mode improves comparative performance on standard tasks by scaling models, adjusting training recipes, and modifying architectures and data. The central artifact is a system together with transparent comparisons on leaderboards and ablation studies that indicate which choices influenced the metric. Success is a higher score at fixed resources or at clearly reported increases in resources. The unit of evidence is a demonstrated score gain supported by controlled ablations. The strength of this mode is that it maps where and how well a method works across tasks and scales. Its limitation is that it remains largely agnostic to mechanism and can confound the effects of scale, data, and compute unless coupled to analyses that target explanation.

**Illustrative papers.** AlexNet (Krizhevsky et al., 2012), ResNet (He et al., 2016), and EfficientNet (Tan & Le, 2019); in multimodal and language, CLIP (Radford et al., 2021), GPT–3 (Brown et al., 2020), and Chinchilla (Hoffmann et al., 2022).

**Mode B: Idealized theory**

This mode derives provable statements in stylized or asymptotic regimes, for example width tending to infinity, infinitesimal step sizes, or noiseless labels. The central artifacts are theorems, bounds, and limiting dynamics, and the objective is logical correctness under explicit assumptions. The unit of evidence is a formal proof, sometimes accompanied by simulation in simplified settings. The strength of this mode is conceptual precision: it produces definitions, constraints, and organizing principles. Its limitation is that observable consequences for finite modern systems are often left implicit, measurable diagnostics are usually out of scope, and external validity remains unknown until tested.

**Illustrative papers.** Spectrally normalized margin bounds (Bartlett et al., 2017), norm-based capacity and implicit regularization (Neyshabur et al., 2015), exact learning dynamics in deep linear networks (Saxe et al., 2014), and benign overfitting in linear regression (Bartlett et al., 2020).

**Takeaway.** These modes answer different questions: *What works?* (benchmarks) and *What must be true under certain assumptions?* (theory). Our proposal in the rest of the paper is to foreground the missing question: *What should we see if a given idea about mechanism is right?*

### 2.1. What can go wrong?

When pursued in isolation these modes can lead to problems, either specific (with direct impact on the current study) or general (with broad impact on the research community):

**Benchmark myopia (Mode A).** Optimizing a single score obscures mechanism and is inconclusive when multiple changes move the metric in offsetting ways. *Consequence:* improvements are hard to attribute; negative or mechanism-revealing results are underreported.

**Transfer gap (Mode B).** Theorems crafted in idealized limits rarely issue observable predictions for real models. *Consequence:* theory informs intuitions but does not guide measurement; experiments cannot falsify the claims.

**Non-cumulative findings (within empirical practice).** Exploratory ablations and robustness checks often document effects without anchoring them to hypotheses or controls predicted to be inert. *Consequence:* results do not compose; later studies cannot easily reuse or contradict prior claims.

**Complexity premium.** Reviewing often treats complexity and scale as a proxy for depth: large models, elaborate training pipelines, and intricate theory are seen as more serious than simple baselines and ideas that isolate a mechanism. *Consequence:* incentives favor "bigger and more complex"

over "cleaner and truer" and discourage small, sharp experiments or proofs that make precise, testable commitments.

**Resource asymmetry.** State-of-the-art (SOTA) incentives, meaning incentives to achieve or surpass the best reported performance on a benchmark or task, disproportionately reward compute- and data-heavy results. This makes it costly to test ideas unless they are bundled with large-scale experiments or elaborate systems. Large labs can distribute work across many people and machines; smaller groups cannot. *Consequence:* publication bias toward scale, underinvestment in mechanism-driven tests, weaker reproducibility for labs without access to large budgets, and a higher chance that simple but sharp ideas never leave the drawing board.

## 3. Ideas First

Now that we have introduced the standard modes of research in neural models alongside possible problems that they face, we can sketch what research looks like when ideas—not just final metrics—do the heavy lifting. In this view, an idea earns weight by making observable commitments (signatures) that experiments can seek and stress-test. Our proposed framework can be summarized as

$$idea \rightarrow signature \rightarrow tailored\ experiment.$$

We treat progress as a chain from *ideas* to *experiments*, with *signatures* as the link. An idea is useful when it makes something *visible*; an experiment is useful when it is *built to see that thing*. Benchmarks remain valuable instruments for external validity and regressions, but *signatures*, not ranks, are our unit of explanation and value. Let's elaborate on our three linked notions of ideas, signatures and experiments.

**What is an idea?** An *idea* is a scope-bearing claim about *how or why* a system behaves, usually established in a *simplified setting* (e.g., single layer, tied weights, infinite width, synthetic data) where analysis or controlled observation is feasible. The idea names the mechanism or constraint, states where it is meant to apply and where it may fail, and issues *observable commitments*. Its value is conceptual and predictive, not benchmark-driven: it makes something precise enough that we can later look for it beyond the toy regime.

**What is a signature?** A *signature* is the concrete way an idea would *show up in a complex/modern model*. It translates the idea's commitments into measurable phenomena—geometry, dynamics, causal responses, invariances, thresholds, or characteristic error patterns—that should be detectable (perhaps approximately or in restricted regimes) when we move from the simplified setting to realistic architectures, scales, and data. A good signature specifies *where*

and *how* to look (layers, training phases, sweeps) and *what trend or boundary* to expect.

**What is a tailored experiment?**  A *tailored experiment* is an empirical test in a complex model that is *designed to observe the signature*. The idea determines the readouts, perturbations, controls, and the region of model/data space where detectability is highest. Success is the clear visibility (or principled absence) of the predicted pattern, not a leaderboard delta. Concretely: (i) define the signature as a statistic/visual you can resolve; (ii) choose instruments that expose it (measurements, ablations/repairs, counterfactuals); (iii) place the measurement where the idea predicts signal (layers/training windows/scale); (iv) sweep the knob the idea says should modulate the signature and include *negative controls* the idea says should not; (v) report the qualitative trend, thresholds, and failures that refine scope.

Modern systems are noisy and heterogeneous; pointwise perfection is the wrong target. We evaluate signatures *in expectation* or as *coarse* patterns (e.g., "usually increasing"), and we allow bounded exceptions. What matters is that the predicted *shape* is visible with reasonable aggregation or replication, not that every unit or datapoint obeys it.

**Why this helps.**  Our position directly combats *benchmark myopia* by shifting the basis of evaluation from marginal score improvements to the explicit detection or falsification of predicted signatures. By requiring that abstract ideas manifest as observable behaviors in finite models, we bridge the *transfer gap*, translating idealized theory into concrete measurements that experiments can use to refine or reject hypotheses. Furthermore, anchoring ablations and robustness checks in specific predictions addresses the problem of *non-cumulative findings*, transforming scattered exploratory results into hypothesis-driven tests that future work can reliably replicate. Emphasizing sharp experiments that make a signature visible regardless of system size counters the *complexity premium*, establishing conceptual clarity and testable commitments as the primary currency of evidence. Finally, by legitimizing modest, carefully targeted experiments as sufficient to substantiate an idea, the framework mitigates *resource asymmetry*, empowering low-compute groups to participate substantively in shaping the field.

## 4. Illustrative Examples

Here we present a list of three well-known works where our approach is present:

### 4.1. Linearized training (NTK)

In the infinite-width limit, training a network is equivalent to kernel gradient flow with a *fixed* Neural Tangent Kernel, yielding a linearized, feature-frozen dynamics (Jacot et al., 2018). This gives a crisp bridge from an idealized regime to modern practice: if the idea carries over at scale, early training of big models should visibly track their NTK linearizations before feature learning takes over.

**Idea:**  In simplified settings (wide limits; squared loss; isotropic inputs), gradient descent follows kernel regression under the NTK fixed at initialization; predictions evolve linearly around the start point (Jacot et al., 2018).
**Signature:** In complex models, during early epochs the network's predictions and loss trajectory closely match those of its NTK-at-init linearization; agreement improves with width and fades as features move.
**Tailored experiment:** Instantiate the linearized predictor and compare it to full training across widths/epochs, reading out prediction/loss alignment and its breakdown. Lee et al. (2019) do exactly this for CNN/ResNet/WRN on CIFAR, directly exposing the early-time linearization signature.

### 4.2. Implicit bias to max-margin

Gradient descent on separable problems implicitly prefers maximum-margin solutions: a phenomenon proved for linear models with exponential-tail losses and extended to homogeneous deep nets (Soudry et al., 2018; Lyu & Li, 2020). The simplified analysis predicts a concrete directional trend that should leave traces in modern representations.

**Idea:** For linearly separable data, GD on logistic/exponential losses drives the weight direction toward the hard-margin SVM solution; analogously holds for homogeneous networks (Soudry et al., 2018; Lyu & Li, 2020).
**Signature:** In complex models, once training error hits zero, the *normalized margin* in penultimate-layer features continues to increase and the classifier direction aligns with a max-margin solution.
**Tailored experiment:**  Track normalized margins and alignment-to-SVM over epochs after interpolation. Lyu & Li (2020) report monotone (smoothed) margin growth and alignment trends on MNIST/CIFAR with MLPs/CNNs, making the max-margin signature visible.

### 4.3. Mixup (heuristic)

Training on convex combinations of inputs and labels encourages approximately linear behavior between examples (Zhang et al., 2018). The heuristic is easy to see in toy 2D settings where decision boundaries straighten along line segments and scales to large vision/speech models, suggesting a clear signature to seek in modern networks.

**Idea:**  In simplified settings, decision regions straighten along line segments between examples and logits vary smoothly with the mixing coefficient (Zhang et al., 2018).
**Signature:** In complex models, along interpolation paths

$x_\lambda = \lambda x_i + (1-\lambda)x_j$, predicted logits for the mixed label vary roughly linearly in $\lambda$; memorization under label noise is reduced and gradients are smoother between samples.
**Tailored experiment:** Probe logits vs. mixing coefficient on real datasets/models and stress-test with corrupted labels; report interpolation-linearity and reduced memorization alongside standard accuracy (Zhang et al., 2018).

## 5. Call to Action: A Field Guide

Now we provide a brief field guide for authors and reviewers that translates our proposal into concrete suggestions for writing and evaluating idea centric work.

### 5.1. For authors

**Specifying the idea.**   *Aim to* (i) state a one or two sentence claim, whether it concerns a concrete method, improvement or a stylized theoretical result; (ii) make the scope explicit in terms of architectures, data regimes, training setups, or the class of models and assumptions considered in the theory; and (iii) mention at least one plausible failure mode or limitation, for example regimes where you do not expect the claim to apply. *Avoid* (i) presenting only a slogan or heuristic with no precise claim; and (ii) letting the idea be defined only via a specific experiment, benchmark gain, or theorem statement without intuitive interpretation and scope.

**Defining signatures.**   *Aim to* (i) introduce a small set of signatures that you reuse throughout the paper; (ii) for each signature, specify the measured or derived quantity, how it is computed, and the expected pattern; and (iii) state regimes where the signature should appear and where it should weaken or disappear. Signatures can be empirical measurements, such as margin distributions or error patterns, or theoretical predictions, such as scaling laws, shapes of learning curves, or stability properties that could be tested. *Avoid* (i) using generic predictions such as "better generalization" without concrete measurements or model level predictions and (ii) introducing many loosely related metrics, lemmas, or bounds whose relation to the idea is unclear.

**Designing tailored experiments.**   *Aim to* (i) make the primary outcome the presence, absence, or strength of your signatures, using empirical measurements or simulations; (ii) use the simplest models and datasets that remain faithful to the stated scope of the idea while plausibly exhibiting the same qualitative patterns as more complex state of the art systems; and (iii) include at least one stress test or boundary case where the idea predicts change, weakening, or failure, for example by relaxing a key assumption or moving outside the intended regime. *Avoid* (i) treating benchmark gains or theorem strength as a substitute for testing signatures and (ii) scaling models and data sizes, or adding layers of technical

assumptions, without a corresponding change in what the idea actually claims about mechanisms or behavior.

### 5.2. For reviewers

**Evaluating the idea.**   *Aim to* (i) judge the clarity, mechanism, and scope of the claim independently of raw performance numbers or formal sophistication; (ii) check that the idea is specific enough that data, simulations, or counterexamples could contradict it; and (iii) reward connections to and tensions with prior empirical and theoretical work, not only incremental gains. *Avoid* (i) rejecting mainly because the paper does not present new state of the art benchmarks or fully general theorems and (ii) demanding large scale experiments when the main contribution is theoretical and/or conceptual, or demanding complete formalization when the stated contribution is mostly empirical.

**Evaluating signatures.**   *Aim to* (i) check that signatures are concrete, measurable, or otherwise operationalizable, and clearly derived from the idea or model; (ii) look for explicit regimes where signatures should appear and for proposed negative controls or assumptions under which they should fail; and (iii) encourage signatures that other groups could realistically test or extend, whether by running new experiments or by sharpening or generalizing the theory. *Avoid* (i) asking primarily for more benchmarks or more theorems when the missing piece is clearer signatures and (ii) accepting vague or post hoc patterns, or technically involved results with no interpretable signature, as sufficient evidence for the idea.

**Evaluating tailored experiments.**   *Aim to* (i) evaluate how well the experiments and simulations align with the stated signatures and scope; (ii) prioritize the informativeness and design of these probes over sheer scale or technical difficulty; and (iii) value reporting of failures, anomalies, and boundary cases that refine the idea or expose limits of the formal model. *Avoid* (i) automatically requesting larger models or datasets when the claim is not about that regime, or stronger and more general theorems when the idea is already well supported by partial results and empirical signatures, and (ii) treating small benchmark deltas or theorem strength alone as the main success criterion for an idea centric paper.

## 6. Hypothetical case study: Investigating topic inertia in LLMs

To illustrate the proposed framework in practice, we present a **hypothetical case study**, inspired by Jia & Diaz-Rodriguez (2025), investigating how large language models (LLMs) maintain topic consistency as context grows.

**The idea.** Since the main components of LLMs are attention

layers, we begin with a hypothesis derived from a simplified theoretical setting: a single-layer self-attention model (Vaswani et al., 2017) with a unified key-query matrix. Theoretical analysis in this idealized regime suggests that as the input sequence length increases, the attention mechanism becomes increasingly biased toward the dominant semantic cluster of the input. We generalize this concept as "Topic Inertia": the probability of an LLM adhering to the topic of the input sequence increases as the input length grows. *Scope:* This claim is grounded in the mechanics of standard self-attention and is not expected to extend to non-attention architectures (e.g., RNNs) or complex agentic workflows involving multi-step reasoning.

**The signature.** We translate this idea into a concrete, observable pattern for modern, deep LLMs. *Predicted Signature:* The semantic similarity (measured via cosine similarity of embeddings) between the input prompt and the generated text should exhibit a generally increasing trend as the number of tokens in the prompt increases. A flat or decreasing trend would falsify the hypothesis.

**The tailored experiment.** To detect this signature, we design a targeted experiment. (i) We select a curated set of 200 documents from a private corpus unknown to public models to rule out memorization effects. (ii) We sample text segments of varying lengths (from 10 to 200 tokens) from these documents and we use them as prompts to query multiple open-weights models: LLaMA 2 (Touvron et al., 2023), GPT-NeoX-20B (Black et al., 2022), MPT-7B (MosaicML, 2023). As a negative control, we also generate with a simple RNN baseline (which lacks the attention mechanism). (iii) We compute the cosine similarity between SBERT embeddings (Reimers & Gurevych, 2019) of the prompt and the generated text for varying prompt lengths.

**Hypothetical results.** In Figure 2 we observe all tested LLMs display the predicted average upward trend in similarity as context length increases, while it is absent in the RNN baseline. We conclude that the signature is visible, supporting the Topic Inertia hypothesis within the defined scope. We explicitly note that these results apply to fixed-size inputs and do not make claims regarding topic drift during long-form generation (increasing output length).

**Hypothetical Contributions.** The hypothetical contributions of this work are: (i) the formalization of the *idea* of "Topic Inertia," demonstrated theoretically in simplified self-attention models; and (ii) the empirical observation of its predicted *signature* in complex LLMs through *tailored experiments*. This demonstrates that the identified mechanism extends beyond theoretical results to accurately describe the behavior of complex Large Language Models.

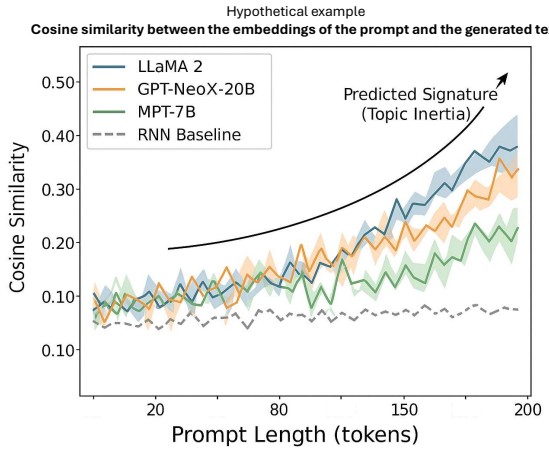

*Figure 2.* The framework on a hypothetical example. The *Idea* (Topic Inertia) predicts a specific *Signature*: an upward trend in embedding similarity as prompt increases. The *Tailored Experiment* confirms this signature in Transformer models, while the negative control (RNN) shows no such trend, validating the mechanism.

### 6.1. Hypothetical Criticisms and the Defense of Scope

If this study were submitted under the current evaluation regime, it would likely face rejection for failing to meet the implicit requirements of benchmark-driven engineering or idealized theory. By anticipating these criticisms, we illustrate how our framework protects scientific value from being discarded due to structural biases.

**The critique of scale and SOTA (Mode A).** A reviewer might object: *"Why stop at 200 tokens? Why evaluate on older open-weights models instead of frontier better systems like GPT-5.1 (OpenAI, 2025) or Gemini 3 (Google DeepMind, 2025)? Does this insight improve perplexity on the LongBench (Bai et al., 2024) leaderboard?"* Under our framework, these demands enforce a **resource threshold** rather than ensuring scientific rigor. If the signature of "Topic Inertia" is clearly resolvable at 200 tokens, scaling the experiment requires massive compute without adding mechanistic insight. Furthermore, demanding evaluations on closed, proprietary APIs introduces reproducibility issues and financial barriers. A tailored experiment maximizes the ratio of insight to compute; demanding scale for scale's sake excludes researchers in low-resource settings and perpetuates "Red AI" without refining the idea.

**The critique of exactness and ablation (Mode B).** A reviewer might also ask: *Why does the trend fluctuate and dip rather than strictly increasing as the theorem suggests? Why did you only test Cosine Similarity and not sweep across all possible embedding metrics?* This reflects a confusion between a **signature** and a mathematical law. Real-world data is noisy; the scientific question is whether the mechanism is visible *in expectation*. The "dips" in the graph represent natural variance, not a falsification of the trend. Similarly, demanding exhaustive ablations across every possible met-

ric is a form of the **complexity premium**. Since the tailored experiment included a negative control that ruled out generic artifacts, additional metric sweeps serve only to increase the authors' workload, not the validity of the claim.

**The "Ideas First" standard.** Ultimately, an "Ideas First" culture changes the burden of proof. It allows authors to state: *"The idea is clearly visible via the predicted signature; optimization and scaling are future work."* If the mechanism is isolated and the signature is robustly detected against controls, the paper has succeeded. The community must learn to accept clear, bounded interesting insights without penalizing them for not being total, SOTA systems.

## 7. Discussion

Our central premise is that the primary unit of scientific value in ML should be the *idea*, rather than a marginal leaderboard gain or an intricate theorem. Currently, the field often inverts this priority, judging work by benchmark numbers or formal guarantees in idealized settings. We argue that this regime is epistemically fragile and structurally biased. An alternative culture centered on ideas and tailored experiments is both scientifically grounded and ethically preferable.

### 7.1. Benchmark Centrism and Its Limits

The dominance of benchmarks encourages the use of complex notation to obfuscate rather than clarify, and a fixation on incremental SOTA results, often obscuring *what is actually being learned* (Lipton & Steinhardt, 2018). Koch & Peterson (2024) describe benchmarking as an "epistemic monoculture" that privileges specific tasks while marginalizing alternative goals.

Empirically, benchmark conclusions are surprisingly fragile. Model superiority often depends more on the specific test set than the method's intrinsic properties (the "Benchmark Lottery") (Dehghani et al., 2021), and small perturbations or variance in seeds can swamp reported gains (Bouthillier et al., 2021; Alzahrani et al., 2024). Furthermore, standard evaluation pipelines can yield misleading conclusions about transferability (Singh et al., 2025a) and ignore critical costs like energy and fairness (Ethayarajh et al., 2020).

Leaderboards are not neutral scoreboards; they are socio-technical artifacts that can entrench power imbalances (Eriksson et al., 2025) and encourage overfitting to the evaluation environment itself (Singh et al., 2025b). Specialized domains face similar issues, where widely used datasets can systematically bias conclusions in drug discovery and scientific ML (Cieplinski et al., 2023; Thiyagalingam et al., 2022). While benchmarks remain indispensable infrastructure (Deng et al., 2009; Wang et al., 2018; Walters, 2023), treating marginal improvements as the primary currency of research is at odds with robust science.

### 7.2. Idealized Theory and Partial Understanding

Theoretical work faces a parallel transfer gap (see Figure 1). Classical frameworks often fail to explain the generalization of deep networks that fit random noise (Zhang et al., 2017). While recent science of deep learning efforts are valuable (Baraniuk et al., 2020; Drori et al., 2022), they frequently rely on stylized models and asymptotic regimes that depart from frontier systems.

Philosophers of science emphasize that laws and theories are typically partial, holding only in circumscribed situations (Cartwright, 1983; da Costa & French, 2003). Scientific understanding often emerges from the interplay of models and exploratory experimentation rather than universal derivation (Giere, 2019; Frigg & Hartmann, 2025). As in physics, where theory must remain aligned to experimental feedback (Ellis & Silk, 2014; Hossenfelder, 2018), mechanistic understanding in ML requires probing trained models directly (Räz & Beisbart, 2024). We advocate for a practice where theory is treated as partial and idealized, and experiments possess enough autonomy to stabilize phenomena independent of a single theoretical framework (Hacking, 1983).

### 7.3. Resource Inequality and the Bias of Current Practice

Demanding large-scale experiments and extensive ablations as a default requirement for publication creates a resource threshold for legitimacy. This exacerbates the "compute divide", concentrating research power in a few well-resourced labs (Ahmed & Wahed, 2020). "Red AI" practices that chase accuracy at any cost worsen environmental impacts and research inequality (Strubell et al., 2019; Schwartz et al., 2020; Xu et al., 2021).

While recommendations for extensive reproducibility checks are well-intentioned (Gundersen et al., 2022; Nature Editorial, 2021), they structurally disadvantage researchers in the Global South and teaching-focused institutions (Mohamed et al., 2020; Chan et al., 2021). Current norms effectively assert that a contribution is only valuable if validated in a regime dominated by a small set of actors (Wu et al., 2022). Positioning idea-centric research as a methodology for "Frugal AI" where systems achieve high impact with minimal resources is therefore critical (UNESCO, 2025). By validating "modest" experiments that clearly isolate a mechanism, the field can broaden participation and mitigate the colonial power relations currently reproduced by AI development (Ayana et al., 2024).

## 7.4. Simplicity and Cumulative Science

Scientific merit should not be conflated with model complexity. Simple algorithms and conceptual proposals are often dismissed as preliminary if they lack large-scale demonstrations, yet simplicity often increases evidential value by isolating the mechanism of interest.

History suggests that major scientific advances emerge from chains of small, incremental contributions (Merton, 1973; Kuhn, 1962). A culture that filters out implementable ideas because they lack heavy engineering risks damaging the field's future. Protecting the ecology of simple ideas allows for a cumulative process where insights can be tested and scaled by others. A reviewing culture that values idea-centric work would reduce the "fixed cost" of participation and prevent the loss of valuable concepts at the submission stage.

## 7.5. Balancing Rigor, Benchmarks, and Idea-Centric Evaluation

We do not argue against theoretical rigor or benchmarks, but for diversifying what counts as evidence. Benchmarks should be viewed as evolving socio-technical artifacts (Hardt, 2025; Eriksson et al., 2025) rather than static scoreboards. Theory should be acknowledged as partial and guided by exploratory experiment (Steinle, 2002).

Ultimately, methodological choices are inseparable from questions of justice (Buolamwini & Gebru, 2018; Noble, 2018). By valuing strong ideas supported by appropriately scaled experiments and negative controls, we can align ML research with the history of successful science while fostering a more equitable and robust community.

# 8. Alternative Views

Our proposal sits within a broader landscape of reasonable disagreement. Many researchers accept that benchmarks, large-scale ablations and strict reproducibility standards are imperfect yet still regard them and their leaderboards as indispensable coordination tools. Similarly, idealized theory and the emerging science of deep learning are seen as the primary route to general principles, with transfer to contemporary systems treated as a long-term goal. We disagree to the extent that this stance treats benchmark gains and idealized theory as self-justifying, rather than requiring a clear link to concrete ideas and model behavior.

A second concern is that simple models and tailored experiments might be less trustworthy than benchmark evaluations, since targeted tests can be unconsciously tuned to confirm a preferred story. From this perspective, strong benchmark improvements or theorems in clean settings are valued because they are harder to manipulate and easier to evaluate consistently. We argue instead that explicit behavioral signatures, and negative controls can make targeted experiments at least as rigorous and reproducible as broad benchmarks.

An alternative view on the role of equity in methodological choice is that resource inequality and the marginalization of small labs are serious social problems but should not shape epistemic standards. If frontier applications require massive compute and complex systems, it is then acceptable that only a few institutions can fully participate, and equity should be addressed through funding and policy rather than by redefining good evidence. We disagree because methodological standards and distributive justice are intertwined, and norms that systematically exclude many researchers are both epistemically and ethically costly.

Another alternative view is that some theoretical research in machine learning should be valued in its own right, even when it is not primarily aimed at explaining the behavior of contemporary models. Such work may advance the mathematics of machine learning or contribute to adjacent fields such as statistics and theoretical computer science. We view this as an important and legitimate agenda. Our aim here is narrower: to argue for a complementary program in the space between idealized theory and benchmark-driven experimentation, where ideas are evaluated through observable signatures in modern systems.

Finally, some see machine learning as closer to engineering than to basic science, where value is measured primarily by deployed systems, competitive performance, and real-world impact. Within this paradigm, theory, benchmarks, and large-scale experimentation are evaluated mainly by their contribution to products and capabilities, and a program that foregrounds ideas and signatures may seem secondary. We disagree because even in engineering-dominated fields, durable progress depends on understanding mechanisms rather than relying only on short-term performance gains.

## Conclusion

We have argued that ML research is currently constrained by a dual fixation on leaderboard dominance and idealized theory, which excludes low-resource researchers, particularly those in the Global South. By adopting the Ideas First framework, the community can shift its focus to understanding behavior. This approach restores the cumulative nature of scientific discovery by validating clear, scope-limited hypotheses through accessible experiments. Ultimately, prioritizing mechanistic clarity over SOTA metrics is not just a methodological correction; it is an ethical imperative that democratizes participation and fosters a more sustainable, scientifically rigorous future for AI.

## Acknowledgments

This work was supported by the Natural Sciences and Engineering Research Council of Canada (NSERC) under grant DGECR-2022-04531. The author thanks the anonymous reviewers, and the session chair for their valuable feedback and insightful suggestions, which greatly improved the quality of this work.

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
