# OpenReview forum: "Position: Ideas Should be the Center of Machine Learning Research"
_ICML.cc/2026/Position_Paper_Track — ICML 2026 Position Paper Track spotlight_

### Official Review · Reviewer_1vym · 2026-02-18

**Significance:** 4
**Argument Clarity:** 4
**Rating:** 5
**Confidence:** 4

**Questions:**

1. There are many ideas that can only be evaluated in large scale in the era of LLM, e.g., scaling law, and perhaps "Vit is better than CNN". Do the authors think the idea-centric framework can help evaluate these ideas?

2. Come back to the budget of reviewing, the idea-centric framework basically needs the reviewer to judge a paper with first-principle thinking, and this does not seem easy for many reviewers. Not to mention that the assessment of an idea is also related to the reviewers' taste. Do you think the mechanism of the reviewing system needs to be changed for the ideal-centric framework?

**Alternative Views Section:**

Yes

**Compliance With Llm Reviewing Policy A Conservative:**

Affirmed.

**Discussion Potential:**

4

**Final Justification:**

I maintain the positive score and recommend acceptance.

**Paper Summary:**

This pape suggest that the machine learning community should embrace an idea-centric framework. The authors first introduce the two current modes, namely benchmark-driven engineering and idealized theory, and argue that they are both suboptimal. They then introduce an idea-centric framework, where the idea is the most important thing. and it can be evaluated by observing the signature in tailored experiments.

**Position:**

Yes

**Position In Title:**

Yes

**Related Work:**

4

**Strengths And Weaknesses:**

Strengths:

1. I will say that the paper is supported by good reasoning and evidence. The authors have given many examples of the two previous modes, and it is clear that benchmark-driven engineering and idealized theory are not perfect.

2. It is very likely to inspire discussion, and the paper is timely. Recently, there have been many discussions in the community, and many people hold the view that engineering is the most important thing in machine learning, especially LLM. So I think this paper will inspire discussion, and I am kind of curious to see how people react to this paper.

Weakness:

1. One thing that can be further elaborate is how to control the budget of reviewing if we advocate this idea-centric framework. At the end of the day, the authors' behavior (which is the policy in reinforcement learning) depends on the reviewers' judgment (the reward). But the idea-centric framework poses more challenges for reviewers, e.g., it is harder to judge the tailored experiments than straightforward benchmark experiments, since they are more case-by-case. I think this position paper would be better if the budget for reviewing is discussed more thoroughly.

**Support:**

3

---

> ### Author Rebuttal · Authors · 2026-03-26
>
> Thank you very much for your comments. See below our reply:
>
> *On the weakness:*
>
> We appreciate this point. An Ideas First framework may make evaluation more case-specific than simply checking standard benchmark tables, but we do not think it necessarily increases reviewing burden overall. Rather, it redirects attention toward the paper’s actual scientific claim: whether it states a clear idea, derives a falsifiable signature, and tests that signature appropriately.
> In fact, benchmark-centered evaluation can itself lead to poor reviewing incentives. When benchmark performance becomes the default proxy for contribution, reviewers, including ourselves at times, may overlook the substance of the paper and focus too heavily on whether the numbers improve over prior work. This can reward marginal gains with limited explanatory value while undervaluing papers whose main contribution is conceptual clarity or a sharp empirical insight.
> Our proposal is therefore not to increase publication volume or reviewing effort, but to improve what gets rewarded. If concise papers built around clear ideas and targeted experiments are valued more highly, while incremental benchmark-driven papers become less competitive, the overall signal-to-noise ratio may improve and some reviewing burden may in fact be reduced.
>
> **Questions**
>
> *1. There are many ideas that can only be evaluated in large scale in the era of LLM, e.g., scaling law, and perhaps "Vit is better than CNN". Do the authors think the idea-centric framework can help evaluate these ideas?*
>
> We agree that some ideas in modern ML can only be meaningfully evaluated at scale. Our framework is not meant to exclude such cases. Rather, it asks that even large-scale studies be organized around a clear claim and its expected observable signature, instead of treating scale alone or aggregate benchmark gains as sufficient evidence. In that sense, Ideas First is compatible with large-scale evaluation, but encourages sharper articulation of what exactly is being tested and why the chosen experiment is the right one.
>
> *2. Come back to the budget of reviewing, the idea-centric framework basically needs the reviewer to judge a paper with first-principle thinking, and this does not seem easy for many reviewers. Not to mention that the assessment of an idea is also related to the reviewers' taste. Do you think the mechanism of the reviewing system needs to be changed for the ideal-centric framework?*
>
> This framework places more weight on judgment than a simple comparison of benchmark numbers. However, we do not think it requires every reviewer to reason from first principles from scratch. A more practical implementation would be to give reviewers a structured set of questions: Is the idea clear? Does the paper derive a falsifiable signature from it? Is the experimental design well matched to that signature? This would make evaluation more explicit and less dependent on implicit taste. So yes, some adjustment in reviewing norms or forms may be helpful, but we do not see our proposal as requiring a wholesale redesign of the reviewing system.

---

> > ### Author Rebuttal · Reviewer_1vym · 2026-04-01
> >
> > Thanks for the detailed response, I maintain the positive score.

---

### Official Review · Reviewer_JVD5 · 2026-03-03

**Significance:** 4
**Argument Clarity:** 3
**Rating:** 4
**Confidence:** 4

**Questions:**

## Question 1

Is it necessary to talk explicitly of neural networks as the only object of interest in contemporary machine learning?

## Question 2

Is the implicit assumption of only modern models being discussed in ML conferences reasonable? What about better understanding fundamental models?

## Question 3

What about exploratory work that does not start from an idea, but rather tries to rationalize what happens in practice in an uncommitted way?

## Question 4

Shouldn't ML conferences be also open to new models that have not yet proven their performance in benchmarks?

## Question 5

How could you bring your framework closer to theoretical work as currently done?

## Question 6

Parallel to question 4, but focused on Mode B: shouldn't we also discuss the extent to which new theoretical frameworks need to be welcomed and encouraged, so that they can reach the maturity necessary to be applicable to modern models?

## Question 7

Shouldn't we use this opportunity to question what proportion of the ML scholarship must be based on heavy compute?

## Question 8

What LLM-supported tools were used in the writing of this paper?

**Alternative Views Section:**

Yes

**Compliance With Llm Reviewing Policy A Conservative:**

Affirmed.

**Discussion Potential:**

4

**Final Justification:**

I remain positive about the paper, but hoping to see the writing revised for reduced redundancy.

**Paper Summary:**

This paper frames current scholarship in machine learning as a dichotomy between benchmark-driven engineering (beating SOTA) and idealized theory distant from modern models (fancy but impractical math), which is representative of sizable portion of papers going around. The authors propose instead to center ML scholarship on ideas that could be tested for what they predict about modern models.

**Position:**

Yes

**Position In Title:**

Yes

**Related Work:**

3

**Strengths And Weaknesses:**

# Strengths

To the extent that it is possible to describe a discipline in which top conferences are now getting more than 30,000 submissions, the authors do a great job outlining two main focal areas. They also advance an alternative that is self-evident when you think about it, but which has not been the norm in peer review. While I am certainly biased for generally agreeing with the perspective presented, I believe that it should be agreed by most in our community that ideas like this one should be put for reflection. Due to the demand push, most of the reviewers in large conferences are quite inexperienced, still working on their PhDs, and they are basically replicating the vices that they have observed in their short time in the community. We need to break this cycle. For that reason, I believe that this position paper should be accepted.

I liked how the authors illustrated their idea with three examples from the literature, and then went on to present a hypothetical study, carry it out at small scale, and conjecture how it would have been received if reviewed in the current ML scholarship culture. I also appreciated the concurrent discussion about equity and how current expectations in scholarship exclude research groups with lesser means and the Global South at large.

To be honest, I look forward to when the reviewing period is over and I can look this paper up and share it around. It is a great contribution, even if not perfect, and for that reason I am picking into a lot of gaps and issues to help polish it in what follows.

# Weaknesses

The writing is good, but not great. Some parts of the text are somewhat abstract, such as the descriptions of Mode A and Mode B, which makes it less accessible for beginner. However, for the sake of opening a debate with experienced ML researchers, the level is adequate. Moreover, the style is repetitive and the modularization makes parts self-contained, which is adequate, but does not flow as easily for reading. You find yourself reading the same ideas followed by the same citations over and over again. At some point you do not expect anything new to appear, and then it does. The problem is that the gold presented towards the end of the paper might be overseen with the current style. Attention spans vanish.

The authors conflate ML research with contemporary neural network architectures. That is an unnecessary simplification in their framing, presented at least in the following excerpts:
- Page 2, Goals, "for modern neural systems": ML is bigger than this; your paper title is about "Machine Learning Research"
- Page 2, Section 2, "neural systems" again
- Page 3, Section 3, narrowing down to "neural models" again

See Question 1 below.

In many ways, the Idea-Centric framing is limited and does not break free from many current biases in ML research:
- What about exploratory work focused on understanding without committing to an idea?
- Why not regarding further investigation of fundamental ("non-modern") models as also important?
- Why not be open to new models, yet to be justified by numbers?
- Why not working backwards from experiments to an idea?

See Questions 2, 3, and 4 below.

From the writing, it seems that the framing proposed by the authors is closer to addressing Mode A (SOTA-oriented research) than Mode B (theory that does not translate to modern systems). While I believe that it represents an advance, it does not connect as well to theory as it does to experimentation. Moreover, the Alternative Views section seems to imply that the combo (Mode A + Mode B) is accepted by most, whereas we could argue that whoever does one may not be keen on the other. For example, consider the excerpts below:
- Page 1, Abstract, "complexity premium" sounds only applicable to Mode A (described in Section 2)by how the sentence ends, not mentioning theory complexity; contrary to how it is discussed later in Section 2, where "intricate theory" is also included. I suggest rewriting that part of the abstract for better clarity.
- Page 3, What is an idea?, when you say "usually established in a simplified setting", it feels as if the whole point is experimenting at a smaller scale (vs. aiming SOTA), which feels more as an alternative to Mode A than to mode B; so it does not address how theory should develop in this "Ideas First" approach.
- Page 3-4, What is a signature?, there is room to discuss Mode B more explicitly around the part saying "when we move from the simplified setting to realistic architectures, scales, and data".
- Page 4, Section 4, expressing all theory as being about formal guarantees is very reductive.
- Page 8, Section 8, in "We disagree to the extent that this stance treats benchmark gains and idealized theory", it sounds as if you are conflating both modes as if they were jointly accepted as standard practice by all ML scholars.

See Question 5 below.

Theory can be quite diverse, and not all theory is disconnected from practice. For example, the structure of linear regions described in theoretical papers has informed the development of algorithms for optimization over trained neural networks. I understand that theory papers are less frequent than engineering papers, so the position paper does cover them somewhat proportionally to their occurrence, but theory also has its own issues. See Question 6 below.

While I understand the turn that the authors take with equity by the Alternative Views section to discuss this as a better distribution of resources, there is a lost opportunity there to question if so much of the ML scholarship should depend on vast amounts of compute power. See Question 7 below.

Finally, I find the modular structure with abundant paragraphs titles a little unusual in general, but not so unusual in my reviewing pile currently. I wonder how much of that is tool-induced. See Question 8 below.

Other minor points and recommendations regarding writing and style:
- Page 1, Reversing the order, remove comma in "to this task, because they".
- Page 2, Change of culture, remove comma in "industrial labs, toward a culture".
- Page 3, Mode B, please define and explain "operational readout" in this context.
- Page 3, Takeaway, final period is unnecessary.
- Page 3, Resource asymmetry: SOTA never defined or explained; I would recommend mentioning earlier what you mean by SOTA and "SOTA incentives".
- Page 4, "Modern systems are [...] paragraph, what is the point of putting this in italics? It sounds as if you are exemplifying what a tailored experiment is, but my understanding is that this is a rephrasing of the idea in discussion.
- Page 4, Why this helps, remove the comma in "regardless of system size, counters".
- Page 6, Specifying the idea, replace colon with semicolon in "result, (ii)" and "theory, and (iii)" (because you have commas separating concepts at different levels here); add semicolon in "claim; and (ii) letting"
- Page 6, Defining signatures, please follow same structure as one item above.
- Page 6, Designing tailored experiments, please follow same structure as two items above.
- Page 6-7, Evaluating the idea, (same as above).
- Page 7, Evaluating signatures, (same as above).
- Page 7, Evaluating tailored experiments, (same as above).
- Page 7, Section 7, put "inspired by Jia & Diaz-Rodriguez (2025)" between commas or em-dashes.
- Page 7, Section 7 up to Section 7.1, casing of paragraph titles is inconsistent and most of them do not match the style in the rest of the paper, where just the first letter is capitalized: change "The [i]dea", "The [s]ignature", "The [t]ailored [e]xperiment", and "Hypotethical [c]ontributions" (note that "Hypothetical results" matches rest of the paper).
- Page 7, The Idea, put W in lowercase here: "layers, [w]e begin".
- Page 7, The critique of scale and SOTA (Mode A), missing opening of double quotes in "of [``] Topic Inertia'' is".
- Page 8, Section 8, either remove the comma in "idealized theory, and the emerging science of deep learning" or add a comma at the end of this excerpt.
- Page 8, Conclusion, this should have a section of its own.

**Support:**

4

---

> ### Author Rebuttal · Authors · 2026-03-26
>
> We appreciate you like our idea and your desire to share it. Thank you for the generous reading, the strong support, and the very thoughtful questions. We are especially glad that you see the paper as a useful intervention in reviewing culture, and we appreciate the care you took in identifying places where the framing can be sharpened.
>
> Let us go one by one on your questions:
>
> *On Q1 and Q2:*
>
> No, the paper is not intended to treat neural networks as the only legitimate object of ML research, nor to imply that only “modern” models matter. Our target is a specific pattern in current flagship ML venues, where evaluation norms are often organized around large neural systems. That is the institutional setting motivating the paper, not a claim about the boundaries of ML as a discipline. The core proposal is broader: an ML contribution should be valued for the clarity of the idea it advances, the observable signature that idea predicts, and the quality of the test used to probe it. That applies just as naturally to classical, probabilistic, kernel, causal, symbolic, or interpretable models, and to work aimed at better understanding foundational models. We can soften the language in the passages you point to so the paper does not read as more narrow than intended.
>
> *On Q3:*
>
> We agree exploratory work is important, and the framework should not be read as excluding it. Some of the most valuable science begins with a puzzle, an anomaly, or a descriptive regularity rather than a sharply stated idea. Our point is that, for such work to accumulate, it eventually helps to move from observation to articulation: what seems to be happening, what signature would make that claim more precise, and what evidence would discriminate among competing explanations. So exploratory work is not outside the framework. It is often an earlier stage that precedes explicit idea formation.
>
> *On Q4:*
>
> Yes, ML conferences should absolutely remain open to new models that have not yet demonstrated top benchmark performance. In fact, that is one of the central motivations for the paper. A new model may matter because it instantiates a genuinely new idea, exposes a distinctive behavioral pattern, or opens a new line of inquiry, even before its performance is fully optimized. A review culture that demands immediate benchmark dominance can suppress exactly the kind of conceptual novelty the field later depends on.
>
> *On Q5-Q6:*
>
> This is a fair challenge. Our argument is not anti-theory. Rather, for the purposes of this paper, we are asking how theoretical work can connect to an Ideas First framework. In that setting, theory is especially valuable when it sharpens understanding in a way that can guide later observation, intervention, approximation, or comparison. An “idea” in theory may take the form of a structural principle, a mechanism, a scaling relation, or a conceptual decomposition, and its “signature” need not be an immediate benchmark prediction. It may instead appear as a qualitative pattern, an invariance, a failure mode, or a comparative prediction that later work can probe empirically or analytically. At the same time, we agree that not all valuable theory in ML needs to be justified in these terms. As we note in our response to reviewer 45PQ, some theoretical research should be valued in its own right, even when it is not primarily aimed at explaining contemporary models. Such work may advance the mathematics of machine learning or contribute to adjacent fields such as statistics and theoretical computer science. We also agree that the same generosity we ask for with new empirical directions should apply to new theoretical frameworks: they should not be required to arrive already mature, complete, and operational on frontier systems in order to matter. Our claim here is therefore narrower: to articulate a complementary space between idealized theory and benchmark-driven experimentation, while still asking how theoretical ideas may eventually develop signatures, bridges, or other forms of contact that make them cumulative.
>
> *On Q7:*
>
> Yes, we agree this is an important implication. The issue is not only equity in the narrow sense of access to resources, but also the deeper question of how much scholarship should depend on heavy compute in the first place. When large scale evaluation becomes the default currency of legitimacy, this can distort what kinds of questions are asked, who can participate, and which forms of evidence are treated as serious. The Ideas First framing is partly meant to counter that by legitimizing contributions whose value lies in sharper hypotheses and better targeted tests, rather than in sheer scale.
>
> *On Q8:*
>
> We used LLM-based tools for editing support, improving writing style, and stress-testing or contrasting our ideas.
>
> Thank you very much for your detailed editing comments. We highly appreciate the care you took in noting them, and we will follow all of them in the revision.

---

> > ### Author Rebuttal · Reviewer_JVD5 · 2026-04-02
> >
> > I am happy with most of the responses, and I hope that the authors consider my points in the final version of their paper.
> >
> > Concerning LLM-supported tools, I still would like to know which tools exactly were used.
> >
> > I do recommend that the authors print and read the paper, or ask a close colleague who has not seen it to do so (likely to be more effective, given that we get saturated of reading our own writing), to see how monotonous the repetition of the same claims get. This paper is a solid 8, but could become a perfect 10 if you "humanize" it.

---

> > > ### Author Response · Authors · 2026-04-02
> > >
> > > Thank you very much for your comments. We appreciate you consider our paper valuable. After reflecting on them and rereading the manuscript, we believe it has three main sources of repetition which might make the paper look "non-human":
> > >
> > > - The Discussion tends to restate the central argument across multiple subsections.
> > >
> > > - The middle of the paper loses momentum, while some of the strongest material in our opinion, such as the hypothetical case study, appears too late.
> > >
> > > - The Introduction is longer than necessary.
> > >
> > > After a careful rereading, we believe the following revisions can address these concerns:
> > >
> > > 1. **Reordering the paper.** We will move the Discussion closer to the end of the manuscript, placing it after the Illustrative Examples and before Alternative Views, and revise the transitions accordingly. This new structure will allow the paper to introduce the Ideas First framework in Section 3, then show through the Illustrative Examples that this mode of research is already present in important parts of the literature. From there, the Call to Action can more naturally motivate the cultural shift we are advocating, followed by the hypothetical example, which shows in concrete terms how such a contribution might look in practice. The Discussion can then serve its proper role near the end of the paper: situating our position in relation to the broader literature on benchmarks, theory, and scientific practice, before closing with Alternative Views.
> > >
> > > 2. **Trimming and refocusing the Discussion.** We agree that the current Discussion is too repetitive. By moving it later in the paper, many of its points will already be established, allowing the section to focus more clearly on connecting our position to related work. In particular, we plan to remove Section 4.3 and cut or revise paragraphs that restate earlier claims or sound overly "non-human".
> > >
> > > 3. **Shortening the Introduction.** We will remove the “Reversing the order” paragraph and fold its main point into “Our position.” While we think that the “Equity and Simplicity” and “Change of Culture” paragraphs are somewhat long for an introduction, we would prefer to retain these ideas in some form because they were among the original motivations for writing this position paper. That said, we will revise them for concision.
> > >
> > > 4. **Clarifying the use of LLM-supported tools.** We will add a short subsection explicitly describing the role of LLM tools in the writing process, including their use for editing, rewriting, and challenging or stress-testing ideas.

---

### Official Review · Reviewer_45PQ · 2026-03-12

**Significance:** 4
**Argument Clarity:** 4
**Rating:** 6
**Confidence:** 4

**Questions:**

What do the authors think the role of benchmark-driven research and theory research should be within the AI and machine learning field? Do the authors think that certain research could justifiable be benchmark-driven or purely theoretical?

**Alternative Views Section:**

Yes

**Compliance With Llm Reviewing Policy A Conservative:**

Affirmed.

**Discussion Potential:**

4

**Final Justification:**

I considered this paper to be quite strong already before the rebuttal. The paper argues its position clearly and is easy to follow. Apart from this, I find the topic to be highly relevant and important. After the rebuttal, the authors addressed my remaining concerns, in particular the one about recognizing advances in mathematics for ML as useful and important on its own, even if it does not immediately result in clear real-world impact.

**Paper Summary:**

In the work, the authors propose to replace benchmark-driven and theory research in AI and machine learning with idea-centric research. Here, the idea is to evaluate concrete hypotheses, derived from conceptual or theoretical considerations, through experiments with a clear signature. The authors define the signature as the interface through which the hypothesis can be falsified. The authors demonstrate issues arising from benchmark-driven research, for example, that different results do not necessarily compose, leaving fragmented findings with unclear connections. Further, the authors also demonstrate issues arising from purely theoretical research, in particular that theory findings do not necessarily transfer to real-world settings or can explain empirical phenomena. The authors then provide examples for idea-centric research from the literature, as well as present a hypothetical case study. Here, the authors develop the idea-centric research from idea to experiment, and discuss which criticisms the study would face in the current landscape.

**Position:**

Yes

**Position In Title:**

Yes

**Related Work:**

3

**Strengths And Weaknesses:**

### Strengths

* One of the biggest strengths of this work is the illustrative examples section, and in particular the hypothetical case study. Both really clarify the position and suggested framework.
* The topic is very timely and important. The authors do a good job of motivating the position, and why a debate on our scientific frameworks is necessary.
* The paper is written exceptionally clearly, and tells a very coherent story.
* While well argued I expect that this position paper might stir serious debate. The alternative view section captures some potential criticisms.

### Weaknesses

I think the paper, and in particular the alternative view section, slightly misrepresents the potential impact and the role of theoretical research. While pure theory might not necessarily explain model behavior, even as a long-term goal, many theoretical questions are scientifically grounded in themselves. This is not necessarily self-justifying, but rather an advancement of an (complementary or adjacent) field: mathematics of machine learning. Some theoretical work can be seen as an advancement of statistics, other theoretical work can be understood as an advancement of theoretical computer science, originating from questions in machine learning. While I do not think that this invalidates the arguments made in the paper, I do think that this is an important nuance to add.

Another thing I want to mention is about the case study, although this is not really a big weakness. Here, the authors claim that the study would likely be rejected by reviewers due to the lack of scale, or proprietary benchmarks. However, I am not convinced that this is true. To give just one example, https://arxiv.org/abs/2310.18988 is a paper accepted at NeurIPS which I would argue falls within the author’s framework, even though not made explicit. Here, the authors come up with the hypothesis that double descent predictably appears even in smaller and less complex models, derived from a conceptual revisit of statistical learning theory and empirical observations made on double descent. The authors then validate the hypotheses in an idealized setting. Other examples of accepted papers that might fit the general framework of idea-centric research: https://arxiv.org/abs/2305.18654, https://arxiv.org/abs/2310.16028, https://arxiv.org/abs/2208.01066. Note that I am not trying to say that idea-centric research is the norm, but I do think that there are many prominent examples of studies which are more-or-less hypothesis driven and evaluated in a controlled and idealized setting, which are also accepted at major conferences. I am happy to hear if the authors think that my examples do not quite fit their idea of idea-centric research.

**Support:**

3

---

> ### Author Rebuttal · Authors · 2026-03-26
>
> Thank you very much for your interesting comments, let us reply to them:
>
> *1. On the Mathematics of Machine Learning*
>
> We appreciate this comment a lot. To be fully honest, we share this concern, and it was one of the points we discussed internally while writing the paper. We agree that mathematics of machine learning should be recognized as a valuable field in its own right, not only as a route toward explaining the behavior of contemporary models. Some theoretical work is better understood as advancing mathematics, statistics, or theoretical computer science through questions originating in ML, and that role deserves clearer recognition than we gave it here. In that sense, we agree there is almost a separate position paper to be written about the value of theory as an autonomous scientific agenda, or even about whether major ML conferences should make more explicit room for that kind of contribution (maybe its own conference track). Our present paper is narrower in scope. Its goal is to articulate a framework for the underdeveloped middle ground between idealized theory and benchmark-driven experimentation, especially in a field where empirical capabilities often advance faster than our foundations. We will revise the paper to make this distinction explicit and to present your point as an important alternative view.
>
> **Alternative views.** We will add to the revised version something in the lines of: "An alternative view is that some theoretical research in machine learning should be valued in its own right, even when it is not primarily aimed at explaining the behavior of contemporary models. Such work may advance the mathematics of machine learning or contribute to adjacent fields such as statistics and theoretical computer science. We view this as an important and legitimate agenda. Our aim here is narrower: to argue for a complementary program in the space between idealized theory and benchmark-driven experimentation, where ideas are evaluated through observable signatures in modern systems."
>
> *2. About the case study.*
>
> Thank you, this is a very helpful point. We agree that the current wording around the case study may be too strong. In particular, saying that such a paper would “likely” be rejected overstates our claim. Your examples are well taken, and we agree that major conferences do accept papers that are hypothesis-driven, mechanism-oriented, and evaluated in controlled or idealized settings.
> Our intention was not to suggest that idea-centric research is absent from top venues, but rather that it is not yet recognized through an explicit evaluative framework that makes clear why this style of contribution is scientifically valuable. As a result, such work can sometimes be assessed through the lenses of the more established alternative modes, which may dilute what is distinctive about it and make its contribution less visibly valued on its own terms. In that sense, the hypothetical case study was meant to illustrate a common review pressure, not to claim that acceptance would be impossible or even unlikely in every case.
> We will revise the text to make this more precise. We also appreciate the concrete papers you mention, since they help illustrate that parts of the framework we advocate already exist in practice, even if not yet named or systematically foregrounded in evaluation.
>
> *Question: What do the authors think the role of benchmark-driven research and theory research should be within the AI and machine learning field? Do the authors think that certain research could justifiable be benchmark-driven or purely theoretical.*
>
> Yes, we do. We see both benchmark-driven and purely theoretical research as legitimate and valuable parts of the field. Our point is not that these modes are unjustified, but that they answer different kinds of questions and should not be treated as sufficient on their own when the goal is to understand model behavior. Benchmark-driven research can show what works, and theory can provide conceptual or mathematical insight, but our argument is that there is also a need for a complementary mode of work that connects ideas to observable signatures in real systems.

---

> > ### Author Rebuttal · Reviewer_45PQ · 2026-04-03
> >
> > The author's rebuttal has adequately addressed my concerns. I appreciate that the authors plan to refine their alternative views section to include a finer distinction regarding mathematics of ML and related subjects. Given that I already considered this work to be quite strong, I decided to raise my score.

---

### Official Review · Reviewer_2YtK · 2026-03-14

**Significance:** 3
**Argument Clarity:** 2
**Rating:** 4
**Confidence:** 4

**Questions:**

I discuss my comments, questions, and weaknesses here.

1. The initiative to put an ideas-first culture seems interesting; however, could the authors mention how this could increase the number of papers being written or accepted? In my opinion, the huge volumes of paper being written are a primary reason the field has become benchmark-driven.

2.  I agree that reviewing these days is more like finding ways to reject a paper, but can authors suggest how reviewers can find an inner threshold? In any case, the scoring is supposed to distinguish between the quality of works, and if all the papers successfully choose a setting and show results in a limited sense, all papers might be scored highly; is that correct?

3. Benchmarking is important in itself, as already mentioned in the paper. However, in my opinion, a lot of asking for additional experiments is only to rule out the concerns a reviewer might have in another setting. For example, if the paper only used one or two datasets, would it be justifiable for the reviewer to ask about some other datasets which they consider are difficult to learn and are still within their hypothesis?

4.  I believe the argument could have been stronger had the authors added the potential fabrication of data as a consequence of benchmarking.

5. Similarly, the reviewing process could be short in ML conferences and asking for additional experiments might not be deliverable for the authors, which could lead to bad scientific practices to get them. In short, I want to say there are many arguments that could have been added to support the authors position.

6. In terms of writing and presentation, I believe the authors could have done a better job.

**Alternative Views Section:**

Yes

**Compliance With Llm Reviewing Policy A Conservative:**

Affirmed.

**Discussion Potential:**

3

**Paper Summary:**

The paper advocates that machine learning research should focus more on the idea rather than benchmarks or theoretical proofs in idealised settings. The paper verbally formalises their position on how the machine learning structure currently is and what must be changed to bring the ideas-first culture. The authors provide examples of papers where their proposed way of research was used, and a call to action is also provided on how to change the research culture in Machine Learning.

**Position:**

Yes

**Position In Title:**

Yes

**Related Work:**

3

**Strengths And Weaknesses:**

The argued position appears timely for the machine learning community, where we see a lot of complaints from the researchers about review processes and from basic sciences researchers who often label ML as empirical science. The arguments, such as enabling researchers in low-resource settings and bridging theory and practice in ML, are well-argued.

I will discuss weaknesses in the questions sections

**Support:**

2

---

> ### Author Rebuttal · Authors · 2026-03-26
>
> Thank you very much for your interesting comments, let us reply one by one:
>
> *1. ...how this could increase the number of papers...the huge volumes of paper being written are a primary reason the field has become benchmark-driven.*
>
> An Ideas First culture would shift attention away from leaderboard gains as the default proxy for contribution and toward a different evaluative standard: whether a paper makes a clear, falsifiable mechanism-level claim and tests it through an observable signature. Under such a standard, incremental benchmark papers with limited explanatory value would likely become less competitive, while concise papers built around sharp ideas and targeted experiments would be recognized as stronger scientific contributions. In that sense, our proposal is not intended to increase publication volume, but to improve the signal-to-noise ratio of what is rewarded, discourage some proliferation of marginal benchmark-driven work, and give greater visibility to insightful, potentially simpler ideas that can make a clear impact.
>
> *2. ...how reviewers can find an inner threshold? ....if all the papers successfully choose a setting and show results in a limited sense, all papers might be scored highly...*
>
> We do not mean that every paper with a limited but successful experiment should receive a high score. Rather, the threshold should be whether the paper makes a clear, falsifiable, scope-aware claim, derives concrete signatures, and tests them with experiments that directly match that claim. This still permits strong differentiation: papers can vary in the precision of the idea, the quality of the signatures, the use of controls or boundary cases, and the overall informativeness of the evidence. Our point is not to remove selectivity, but to shift it away from benchmark gains and scale alone toward clearer scientific commitments and better-matched evidence, more in line with the natural sciences, where work is valued for isolating mechanisms, testing falsifiable hypotheses, and clarifying the conditions under which claims hold.
>
> *3. Benchmarking is important in itself ...a lot of asking for additional experiments is only to rule out the concerns a reviewer might have in another setting...*
>
> We agree that requesting additional datasets can be justified when they are genuinely diagnostic of the hypothesis, for example by probing a stress case, boundary condition, or regime where the predicted signature should change. Under our framework, however, such requests should follow from the logic of the claim, not from a general expectation of broader benchmark coverage. The key question is whether the added dataset sharpens the test of the idea, rather than simply adding another result table.
>
> *4-5. ...potential fabrication of data as a consequence of benchmarking ...and asking for additional experiments might not be deliverable for the authors, which could lead to bad scientific practices.*
>
> We definitely agree. In the revision, we plan to add a short discussion in Section 4.1 noting that benchmark-centered incentives, especially under short rebuttal timelines, can create pressure for rushed experimentation and poor scientific practice. This fits naturally with our existing argument that treating marginal benchmark gains as the main currency of value can distort both evaluation and research behavior. We also plan to extend Section 4.4 to clarify that expectations for extensive additional experiments do not only disadvantage lower-resource groups, but may also incentivize low-quality last-minute experimentation. We will also make the practical implication more explicit in Section 6.2 for reviewers: requests for additional experiments/ablations should be driven by whether they are genuinely diagnostic of the paper’s central claim.
>
> *5. ...there are many arguments that could have been added to support the authors position.*
>
> This is, in part, precisely why we wrote the paper as a position paper: not to offer an exhaustive account of all the failures of current practice, but to articulate a general perspective, develop several representative arguments in its favor, and illustrate what better practice could look like. Along those lines, we discuss multiple motivations for an ideas-first culture, and also provide concrete examples, a hypothetical review scenario, and practical guidance for reviewers. We intend the paper as a constructive starting point for a broader conversation, rather than as a complete catalogue of every supporting argument. At the same time, we found your suggestions very helpful: the additional arguments you raise are strong, and we agree they would further strengthen the manuscript. We will incorporate these points in the revision.
>
> 6. ...the authors could have done a better job.
>
> We will consider targeted revisions to improve clarity of presentation, especially around the statement of the main proposal and the reviewer-facing takeaways.

---

> > ### Author Rebuttal · Reviewer_2YtK · 2026-04-04
> >
> > Thanks for the rebuttal. I understand the key motivation of the paper and that the idea was not to provide an exhaustive list of failures, but I still feel there could have been more points to discuss in the paper, and some discussions on the grey area. Keeping that in mind, I'm keeping my original evaluation, which is already positive, and I hope the final version leads to fruitful discussions at the conference.

---

### Decision · Program_Chairs · 2026-04-30

**Decision:**

Accept (spotlight)

**Comment:**

The paper presents the position that "ideas" (not benchmark-driven leaderboards or theorems that are presented in idealized and non-transferable conditions) should be the center of research in machine learning.
All reviewers are positive about the paper, noting that the position is timely, important, well-motivated and well-substantiated, and likely to spark significant debate.
During the rebuttal, the authors discussed various issues and made several clarifications, including that their proposed framework is not intended to lead to accepting more papers, nor to compromise the ability to be selective and rank papers in terms of quality, but the reviewing criteria would need to change accordingly.
Some reviewers noted that the writing quality can be improved (there is repetition and the organization can be improved) and that more arguments can be added to the paper. The authors have been receptive to these concerns and have discussed changes to the paper in light of them. The authors also clarified that their position is not that benchmarks should not be used nor that theory should not be encouraged. But that these should be used as a means towards testing a hypothesis, rather than as an end in itself. In discussions with Reviewer 45PQ and Reviewer JVD5, the authors agreed to revise the paper to add an important alternative view that argues for value in theoretical work as an autonomous scientific agenda (not only as a means towards testing a hypothesis about model behaviours).


I recommend acceptance as the paper articulates a clear and well-supported position about a timely and important topic and is likely to spark interesting discussions in the community.